# Spatial distribution, and predictors of late initiation of first antenatal care visit in Ethiopia: Spatial and multilevel analysis

Gossa Fetene Abebe[1]*, Anteneh Messele Birhanu[2], Dereje Alemayehu[3], Desalegn Girma[1], Ashenafi Assefa Berchedi[4], Yilkal Negesse[5]

1 Department of Midwifery, College of Medicine and Health Sciences, Mizan-Tepi University, Mizan-Teferi, Ethiopia, 2 School of Medicine, College of Medicine and Health Sciences, Mizan-Tepi University, Mizan-Teferi, Ethiopia, 3 School of Public Health, College of Medicine and Health Sciences, Mizan-Tepi University, Mizan-Teferi, Ethiopia, 4 Department of Nursing, College of Medicine and Health Sciences, Mizan-Tepi University, Mizan-Teferi, Ethiopia, 5 Department of Public Health, College of Health Sciences, Debre Markos University, Debre Markos, Ethiopia

* Feteneg2119@gmail.com

**Data Availability Statement:** All the data underlying the results presented in the study are

## Abstract

### Background

Despite the proven benefit of early initiation of first antenatal care visits as a means to achieve good maternal and neonatal health outcomes through early detection and prevention of risks during pregnancy, shreds of evidence showed that most of the women in Ethiopia start their ANC visits lately.

### Objective

To determine the spatial distribution and predictors of late initiation of first antenatal care visits among reproductive-age women in Ethiopia.

### Method

The 2019 Ethiopian Mini Demographic and Health Survey (EMDHS) data were used. A total weighted sample of 2,935 reproductive-age women who gave birth in the five years preceding the survey and who had antenatal care visits for their last child was included. To check the nature of the distribution of late initiation of ANC visits, the global Moran's I statistics were applied. Gettis-OrdGi statistics and spatial interpolation using the Ordinary Kriging method were done to identify the spatial locations and to predict unknown locations of late initiation of first ANC visits, respectively. For the predictors, a multilevel mixed-effect logistic regression model was applied. Finally, statistical significance was declared at a p-value < 0.05.

### Results

The prevalence of late initiation of first ANC visits in Ethiopia was 62.6%. The spatial analysis showed that the late initiation of first ANC visits significantly varied across regions of

publicly available from the Measure DHS website: http://www.dhsprogram.com.

**Funding:** The author(s) received no specific funding for this work.

**Competing interests:** The authors have declared that no competing interests exist.

**Abbreviations:** ANC, Antenatal Care; AOR, Adjusted odds ratio; CI, Confidence Interval; HIV, Human Immunodeficiency Virus; SNNPR, South Nation Nationality Peoples Region; WHO, World Health Organization.

Ethiopia. The spatial interpolation predicted the highest rates of late initiation of first ANC visits in the eastern SNNPRs, southern and western Oromia, and some parts of the Somalia region. Being rural residents, attending higher education, having medium wealth status, richer wealth status, richest wealth status, having ≥ 5 family size, a household headed by male, living in SNNPRs, and Oromia regions were significant predictors of late initiation of first ANC visits.

## Conclusion

A clustered pattern of areas with high rates of late initiation of the first ANC visit was detected in Ethiopia. Public health intervention targeting the identified hotspot areas, and women's empowerment would decrease the late start of the first ANC visit. Furthermore, the identified predictors should be underscored when designing new policies and strategies.

## Introduction

Maternal and neonatal health has gained a global health priority in the past decades, exemplified by its adoption as the fourth and fifth Millennium Development Goals (MDG) [1, 2] and continued in the third Sustainable Development Goals (SDG) [3]. Nevertheless, the goal to reduce maternal mortality by 75% in the year 2015 was not successful and lagged behind its target among all [1]. Regardless of the 2015 commitment, about 295,000 women died during and following pregnancy and childbirth in 2017 [4] globally. Sub-Saharan Africa and Southern Asia shoulder the highest burden (86%) of this estimated global maternal deaths, of which Sub-Saharan Africa alone accounted for two-thirds of maternal deaths [4], where the health systems are poor in access and minimum health services utilization [5]. The maternal mortality ratio (MMR) in Ethiopia is 412 maternal deaths per 100,000 live births. Also, globally around 2 million stillbirths [6] in the year 2019 and 2.4 million neonatal deaths [7] in the year 2020 were reported. Of which the highest numbers were contributed by Sub-Saharan Africa including Ethiopia and Southern Asia [6, 7].

The provision of maternal healthcare services which is sustainably available, accessible, and of good quality for all pregnant women at all levels is very important to maximize the health status of the mother and her newborn baby [8–10]. Antenatal care is one of the proven maternal care services strategies for reducing maternal and neonatal mortality directly by the identification and treatment of pregnancy-related illness, or indirectly by detection of women at risk of complications of delivery and counseling them to deliver in an appropriately equipped health facility [11]. Timely (initiated at the first trimester of pregnancy) [12, 13], frequent (four or more visits) [12, 14, 15], and adequate (with proper contents) [12, 14, 15] antenatal care services provided for the mother during perinatal period decrease the risk of complication and death for the mother as well as the unborn fetus [9] and improve the use of subsequent maternal health services. The time at which the first antenatal care visit is initiated has a great advantage to maintain optimal health impacts for both woman and their unborn fetus [11]. According to the 2016 World Health Organization (WHO) antenatal care recommendation for positive pregnancy outcomes, the first antenatal care visits should be within the first trimester (early) [11]. However, the magnitude of early antenatal care visits is very low (24%) in developing countries as compared to developed countries (81.9%) [16]. Also, shreds of evidence noted that many pregnant women attend their first ANC visits late (after the first

trimester) [17–19]. A systematic review study done in Ethiopia by Tesfaye G. et al. identified that the majority (64%) of pregnant women start their first ANC visits late [20].

Early initiation of ANC visits is one of the pillar components of ANC services that aimed to have baseline information on the general well-being of the pregnant women, accurate gestational age ascertainment, screening of pre-existing problems of the women such as human immunodeficiency virus (HIV), syphilis, hepatitis, malaria, anemia, and other chronic medical disorders, and early detection of complications arising during pregnancy [11, 16]. Timely screening and providing appropriate therapy for HIV and syphilis help to halt mother-to-fetus transmission. If the mothers are left untreated early, a 70–100% probability of transmitting the infection to their unborn fetus, and one-third of pregnancies will end up with stillbirth [21, 22].

Moreover, early initiation of ANC is a good entry point to discuss birth preparedness and complication readiness plan and to augment awareness of its sign and symptoms between pregnant women and health care providers [17]. It also creates the opportunity to provide immunizations against tetanus, supplementation of iron and folic acid to prevent anemia and neural tube defect, counseling on nutrition, and malaria and worms prophylactic treatments [11]. Early initiation of antenatal care services also has a positive impact on the decrement of poor perinatal outcomes such as low birth weight, preterm birth, and jaundice [23, 24]. In summary, early ANC visits aim to screen complications or predictors for the occurrence of complications which enable timely interventions to handle the negative impacts of such complications on a pregnant woman and unborn fetus [25].

To improve the uptake of ANC services early, the government of Ethiopia in collaboration with other non-governmental organizations has implemented different initiatives that expand access such as primary health care expansion, health extension programs, and charge-free maternal health services [26–30]. Despite, all these efforts tried, in Ethiopia, the timely initiation of first ANC visits within the first twelve weeks as recommended by WHO [11] is still low, and most pregnant women start their first ANC visits late [20]. As reported in the Ethiopian Demographic and Health Survey (EDHS) 2016, only 20% of pregnant women started their first ANC visit in the first trimester of pregnancy [31] as recommended by WHO [11]. In other fragmented studies conducted in different parts of Ethiopia, the delayed initiation of first ANC visits ranges from 33.3% [32] to 85.56% [33]. This indicated that there are factors that became bottlenecks for the increment of early initiation of first ANC visits and need further investigation.

Previously studies employed in Ethiopia identified that late initiation of first ANC visits was deterred by different predictors like place of residence [20, 34, 35], maternal educational status [20, 34, 35], husband education status [36], maternal occupation [37], maternal age [20, 35], marital status, household wealth-income [20, 35], parity [20, 34], partner involvement [20], pregnancy intention [37], poor knowledge about ANC [37], unintended current pregnancy [20, 36], being advised before starting antenatal care visit [36], exposure to mass media [35], pregnancy complication [20], having a history of abortion or stillbirth [38], covered by health insurance, distance from health facilities [35], and regions [34].

However, those previous studies in Ethiopia were conducted in specific areas with small sample sizes, mainly facility-based, and were not nationally representative. Moreover, the previous studies primarily focused on individual-level factors with little attention given to community-level factors, which could undermine the importance of considering contextual factors when designing appropriate maternal health services strategies. Also, the current spatial distributions and hotspot areas of high rates of delayed initiation of the first ANC visit are not determined. While, a spatial analysis study would give information about hotspot areas for prioritizing, resource allocation, and forecasting of healthcare utilization trends. Therefore, for

bridging all those gaps, this study aimed to determine the spatial distributions, and individual and community-level predictors of late initiation of the first ANC visit among reproductive-age women based on the nationally representative recent data (2019 EMDHS). The findings of this study will be important to point out the hotspot areas and the predictors impeding the late initiation of first ANC visits as a result respective programs are made more consistently towards the identified hotspot areas and predictors. It will also have a paramount benefit to showing country-level figures and screening out modified and persistent factors, which in turn minimize the late initiation of the first ANC visits, and achievement of SDG #3 of reducing maternal and neonatal mortality by 2030.

## Method and materials

### Study setting, period, and design

The present study was conducted based on the 2019 Ethiopia Mini Demographic and Health Survey (EMDHS) data, collected from 21st March/2019 to 28th June/2019. Ethiopia is found in the horn of Africa and located between 3˚–15˚ North latitude and 33˚–48˚ East longitudes. It has nine regions and two city administrations. The EDHS is a nationally representative cross-sectional survey done in all nine regions and two city administrations every five years. In between the standard EDHS, two Mini Demographic and Health Surveys (EMDHS) have been employed in the year 2014 and 2019.

### Data source and extraction

After permission was secured through an online request, the survey and geographic location data (longitude and latitude coordinate) were obtained from the Measure Demographic and Health Surveys (DHS) website (http://www.dhsprogram.com/). The outcome variable late initiation of the first antenatal care visit and its individual and community level variables were extracted thoroughly.

### The population of the study

All reproductive age women (15–49 years) who gave birth in the five years preceding the survey and who had at least one ANC visit for their last child all over Ethiopia were the source population, whereas women who gave birth in the five years preceding the survey and who had at least one ANC visit for their last child and lived in the selected enumeration areas (EAs) were the study populations.

### Eligibility criteria

All reproductive-age women found in the selected clusters at least one night before the data collection period were included, whereas, women whose geographical locations were not available at the global positioning system (GPS) were detered from this study.

### Sampling technique and sample size

In the 2019 EMDHS, a stratified two-stage sampling technique has been used to collect the data. In the first stage, stratification was done by region, and then each region was stratified as urban and rural. A total of 305 (94 urban, and 211 rural) enumeration areas (EAs) were selected using probability proportional to EA size in the first stage. In the second stage, households were selected proportionally from each EA by using a systematic sampling method. Finally, a total of 2,935 weighted samples of reproductive-age women were included in this study. The detailed method of data collection is available in the DHS database [39].

### Variables of the study

**Outcome variable.** The outcome variable of the study was late initiation of the first ANC visit which was a binary outcome and classified as "late" if a woman attend her first ANC visit after the first 12 weeks of gestation, and "early" if she attends the first ANC visit within the first 12 weeks of gestation [11].

**Independent variables.** The independent variables of the study were grouped as individual and community-level variables. Maternal age, marital status, religion, educational status of women, household wealth status, parity, preceding birth interval, the contraceptive method used, sex of household head, and the family size were included in the individual level predictors. Whereas, the place of residence and region of the study participants were considered community-level predictors.

### Data management and statistical analysis

Once the data were extracted from the birth recorded data set of the EMDHS, data cleaning and recoding were done using STATA version 15. To restore the representativeness of the survey and get reliable statistical estimates, the data were weighted for probability sampling and non-response using sample weight. Then descriptive, spatial, and multilevel analyses were carried out.

### Spatial analysis

Microsoft Excel, STATA version 15, and ArcGIS 10.3 software were used to determine the geographic distribution of the late initiation of the first ANC visit. The weighted frequency of the outcome variable with cluster number and geographic coordinate data were merged in STATA version 15. To make ready data for ArcGIS 10.3 for spatial analysis, the data were exported to CSV delimited format.

### Spatial autocorrelation analysis

The spatial autocorrelation (Global Moran's I) statistics were done to determine whether the patterns of late initiation of the first ANC visit were randomly distributed, or not. The Global Moran's I values ranging from − 1 to + 1 were used to identify whether the patterns of late initiation of the first ANC visit were dispersed, clustered, or random in the study area [40, 41]. The statistical value close to– 1 depicts the spatial distribution of late initiation of ANC visit was dispersed, a statistic value close to 0 depicts the spatial distribution of late initiation of first ANC visit was randomly distributed, and a statistic close value + 1 depicts the spatial distribution of late initiation of first ANC visit was clustered.

### Hot spot analysis (Gettis-OrdGi* statistic)

The Gettis-OrdGi* statistic was employed to identify areas that have significantly high hotspot areas [42]. The $Z$-score and $P$-value were computed to determine the statistical significance of clusters. The statistical output having high GI* depicts "hotspot" areas (high rates of late initiation of first ANC visit), whereas having low GI* depicts "cold spot" areas (low rates of late initiation of first ANC visit).

### Spatial interpolation or prediction

To predict the burden of certain events for unsampled areas based on the sampled areas, spatial interpolation was employed. In Ethiopia Mini Demographic and Health Survey 2019, a total of 305 enumeration areas were selected to take a sample that was believed to be

representative of the country. Therefore, in this study to predict the rates of late initiation of first ANC visits for the unobserved areas of Ethiopia, the Ordinary Kriging prediction method was employed based on the sampled clusters.

## Multilevel analysis

Due to the sampling technique used in EMDHS (Multistage stratified cluster sampling), two levels of data hierarchy were considered. Level one women were nested within the household, then the households were nested at the next higher level (enumeration areas). The outcome variable was represented by $Y_{ij} = \begin{cases} \textit{Late initiation of first ANC visit} \\ \textit{Early initiation of first ANC visit} \end{cases}$, the category is dichotomous. Therefore, the multilevel mixed-effect logistic regression model was fitted to identify the predictors impending the late initiation of the first ANC visit at each level. For the multi-level logistic regression, four models were fitted. The null model, a model without predictors, was fitted to determine the extent of cluster variation in the late initiation of first ANC visits. Then, the second model with individual-level predictors alone, the third model with community-level predictors, and lastly, the fourth model was fitted with both individual and community-level predictors. Both bivariable and multivariable multi-level logistic regression analyses were computed. Multicollinearity between independent predictors was detected using the variance inflation factor. In the bivariable two-level binary logistic regression analysis, variables that have a $P$-value of $\leq 0.25$ were candidates for multivariable multilevel logistic regression analysis. Then, variables in multilevel multivariable logistic regression were declared to be statistically significant at a $P$-value of $< 0.05$. The fitted model was compared based on log likely hood ratio (LLR) and deviance (-2LLR). A model with the high LLR and low -2LLR value was selected as best fitted model and all interpretations and inferences were made based on this model. Intraclass correlation coefficient (ICC), median odds ratio (MOR), and proportional change in variance (PCV) statistics were computed to determine variations of late initiation of the first ANC visit across clusters. The variation within clusters and between clusters was detected using ICC. The PCV depicts the total variation of late initiation of the first ANC visit at the individual- and community-level predictors in each model. The MOR determines the MOR of late initiation of first ANC visit at the high-risk cluster (clusters having high rates of late initiation of first ANC visit) and low-risk cluster (clusters having low rates of late initiation of first ANC visit) when we randomly pick two women during data collection from two clusters. These three measurements were computed using the following formula;

$$\text{ICC} = v_i/(v_i + \pi^2/3) \sim \frac{Vi}{Vi + 3.29},$$

where $V_i$ = between cluster (community) variances and $\pi2/3$ = within-cluster (community) variance [43].

$$\text{PCV} = \frac{Vi - Vy}{Vi},$$

where Vi = variances of the null model, where Vy = variance of the model with more terms [43].

$$\text{MOR} = exp.[\sqrt{2 \times Vz} \times 0.6745] \sim exp.[0.95\sqrt{Vz}]$$

where Vz = variance at the community level [43].

## Ethical consideration

Ethical approval was obtained from the measure Demographic Health Survey (DHS) after filling out the requesting form for accessing the data. The data used in this study are freely available, aggregated secondary data that didn't contain any personal identifiers that can be linked to the study participants (http://www.dhsprogram.com). The requested data were used strictly anonymous and served only for the study purpose. The full information about the ethical issue was available in the EMDHS-2019 report.

## Result

### Background characteristics of the study participants

A total weighted sample of 2,935 reproductive-age women was included. The overall magnitude of late initiation of the first antenatal care visit was 62.6% (95% CI: 59.8–65.3). More than half (54.8%) of the study participants were in the age range of 20–29 years old. Of the total study participants, 2,052 (70.2%) were urban dwellers, 2,735 (93.6) were married, and 1,282 (43.9%) didn't attend formal education (Table 1). The highest rates of late initiation of the first ANC visit were reported from the SNNPRs (71.2%) followed by the Oromia regional state (66.8%) (Fig 1).

**Table 1. Late initiation of first antenatal care visits by sociodemographic characteristics of reproductive-age women in Ethiopia, 2019.**

| Variable | Category | Weighted frequency (%) | Late initiation of first ANC | |
|---|---|---|---|---|
| | | | No (%) | Yes (%) |
| Women age (years) | 15–19 | 153 (5.4) | 48 (4.5) | 105 (5.9) |
| | 20–29 | 1,553 (54.8) | 605 (56.6) | 948 (53.8) |
| | 30–39 | 1,013 (35.8) | 381 (35.6) | 632 (35.8) |
| | 40–49 | 113 (4.0) | 34 (3.2) | 79 (4.5) |
| Residence | Urban | 871 (29.8) | 445 (40.7) | 426 (23.3) |
| | Rural | 2,052 (70.2) | 647 (59.3) | 1405 (76.7) |
| Marital status | Married | 2,735 (93.6) | 1,024 (93.7) | 1,711 (93.5) |
| | Unmarried | 188 (6.4) | 69 (6.3) | 119 (6.5) |
| Religion | Orthodox | 1,223 (41.8) | 518 (47.4) | 705 (38.5) |
| | Muslim | 876 (30.0) | 318 (29.2) | 557 (30.4) |
| | Protestant | 793 (27.1) | 246 (22.5) | 547 (29.9) |
| | Others[c] | 31 (1.1) | 10 (0.9) | 22 (1.2) |
| Maternal educational status | No education | 1,282 (43.9) | 404 (37.0) | 878 (48.0) |
| | Primary | 1,153 (39.5) | 453 (41.4) | 701 (38.3) |
| | Secondary | 335 (11.5) | 130 (11.9) | 205 (11.2) |
| | Higher | 152 (5.2) | 105 (9.6) | 47 (2.6) |
| Household wealth status | Poorest | 399 (13.7) | 101 (9.3) | 298 (16.3) |
| | Poorer | 587 (20.1) | 162 (14.8) | 425 (23.2) |
| | Middle | 589 (20.1) | 181 (16.6) | 407 (22.2) |
| | Richer | 578 (19.8) | 220 (20.1) | 358 (19.6) |
| | Richest | 770 (26.3) | 428 (39.2) | 343 (18.7) |
| Sex of household head | Female | 385 (13.2) | 156 (14.3) | 229 (12.5) |
| | Male | 2,538 (86.8) | 936 (85.7) | 1601 (87.5) |

ANC: Antenatal care,

[c] catholic or traditional religion follower

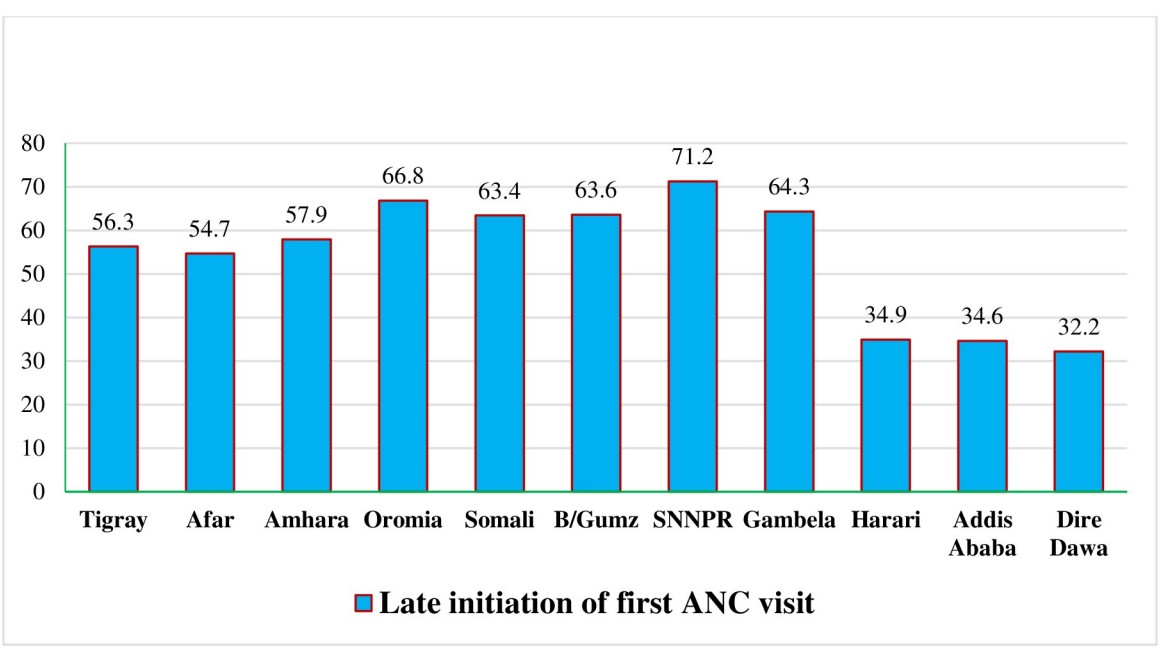

**Fig 1. Regional prevalence of late initiation of first ANC visit in Ethiopia, 2019.** SNNPR: South Nation Nationality peoples region, B/Gumz: Benshangul-Gunz region.

## Maternal health and reproductive-related characteristics of the study participants

More than two-thirds (63.3%) of the households have had ≥ 5 family members and nearly three fourth (76.4%) were multiparous. The majority (62.2%) of the women had given birth more than two times, 57.5% of the woman have had a birth spacing of 36 months and above, and 50% of the women didn't use contraception (Table 2).

## Spatial analysis

### Spatial autocorrelation analysis of late initiation of first ANC visit

The global spatial autocorrelation analysis identified that the spatial distribution of late initiation of the first ANC visit was significantly varied across regions in the country with Global Moran's Index values of 0.87 ($P < 0.001$) (Fig 2). Thus, the global Moran's I is significant and greater than zero, we conclude that the distribution of delayed ANC visits was non-random (clustered).

### Hot spot analysis of late initiation of first ANC visit

The spatial distribution of late initiation of the first ANC visit was clustered in some parts of Ethiopia. Based on the Gettis-OrdGi statistical analysis, the hotspot areas (highest rates of late initiation of first ANC visit) were found in the eastern parts of SNNPR, eastern and central parts of Tigray, central Amhara, western Oromia, and a few places in the Somalia and Afar regions. Whereas, the majority of the cold spot areas (low rate of late initiation of first ANC visit) were found in Dire Dawa, Harari, and Gambella regions (Fig 3).

**Table 2. Late initiation of first antenatal care visit by maternal health and reproductive related characteristics of women in Ethiopia, 2019.**

| Variable | Category | Weighted frequency (%) | Late initiation of first ANC visit | |
|---|---|---|---|---|
| | | | No (%) | Yes (%) |
| Parity | Primiparous | 689 (23.6) | 299 (27.4) | 390 (21.3) |
| | Multiparous | 2,234 (76.4) | 793 (72.6) | 1,441 (78.7) |
| Birth order | First | 421 (19.6) | 154 (20) | 267 (19.4) |
| | Second | 391 (18.2) | 121 (15.6) | 270 (19.6) |
| | Third and above | 1,336 (62.2) | 497 (64.4) | 839 (61) |
| Preceding birth interval (months) | < 24 | 139 (8.1) | 55 (8.9) | 84 (7.6) |
| | 24–36 | 592 (34.4) | 203 (32.9) | 389 (35.3) |
| | >36 | 988 (57.5) | 360 (58.2) | 629 (57.1) |
| Contraceptive method used | No-method | 1,463 (50) | 489 (44.7) | 974 (53.2) |
| | Modern | 1,444 (49.4) | 600 (54.9) | 845 (46.1) |
| | Traditional | 16 (0.5) | 4 (0.3) | 12 (0.7) |
| Family member | ≤ 4 | 1,074 (36.7) | 459 (42) | 615 (33.6) |
| | ≥ 5 | 1,849 (63.3) | 633 (58) | 1216 (66.4) |
| Number of antenatal care | Once | 130 (4.5) | 15 (1.3) | 116 (6.3) |
| | Two times | 293 (10) | 47 (4.3) | 247 (13.5) |
| | Three times | 801 (27.4) | 196 (18) | 605 (33) |
| | Four and above | 1,698 (58.1) | 835 (76.4) | 864 (47.2) |

ANC; Antenatal care

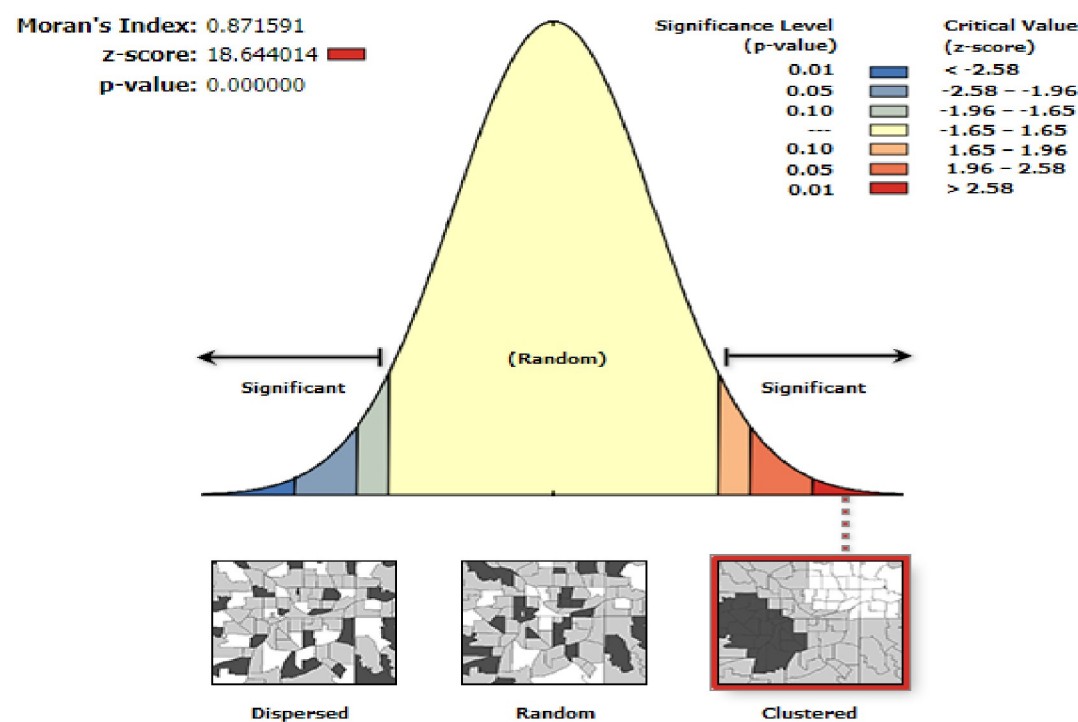

**Fig 2. Spatial autocorrelation (Moran I) of late initiation of first ANC in Ethiopia, 2019.**

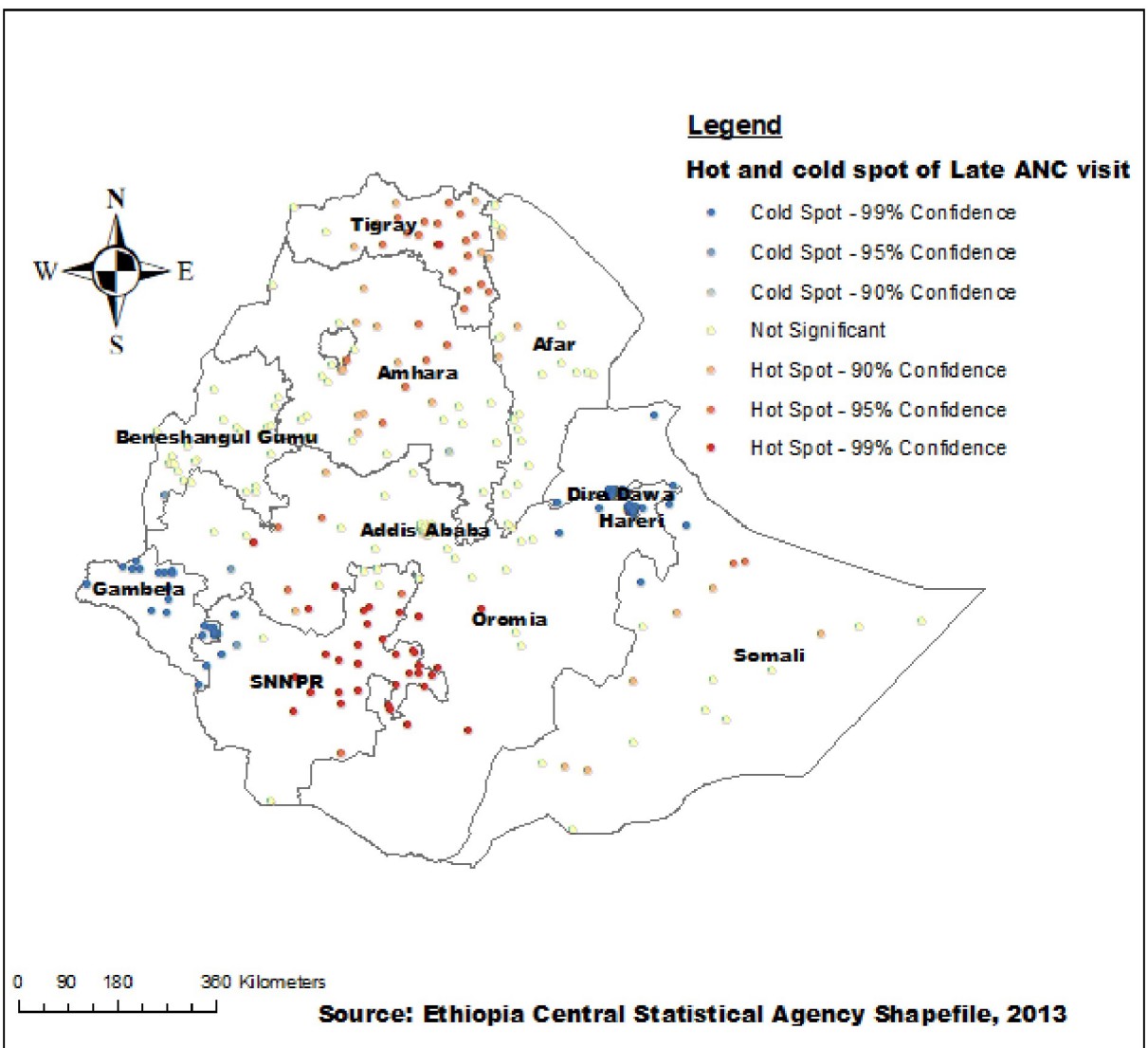

**Fig 3. Hot and cold spot analysis of late initiation of first ANC visits in Ethiopia, 2019.**

### Ordinary Kriging interpolation or prediction of late initiation of first ANC visit

To determine the occurrence of late initiation of first ANC visits across the unsampled sites, Ordinary Kriging interpolation was employed. Based on the EMDHS 2019 data, the highest magnitude of late initiation of first ANC visits among reproductive-age women was predicted in the eastern SNNPR, southern and western Oromia, and a few places in the southern, and western parts of the Somalia region. Whereas, the lowest magnitude of late initiation of first ANC visits was predicted in Gambella, Dire Dawa, Harari, western parts of SNNPR, and northern parts of the Somalia region (Fig 4).

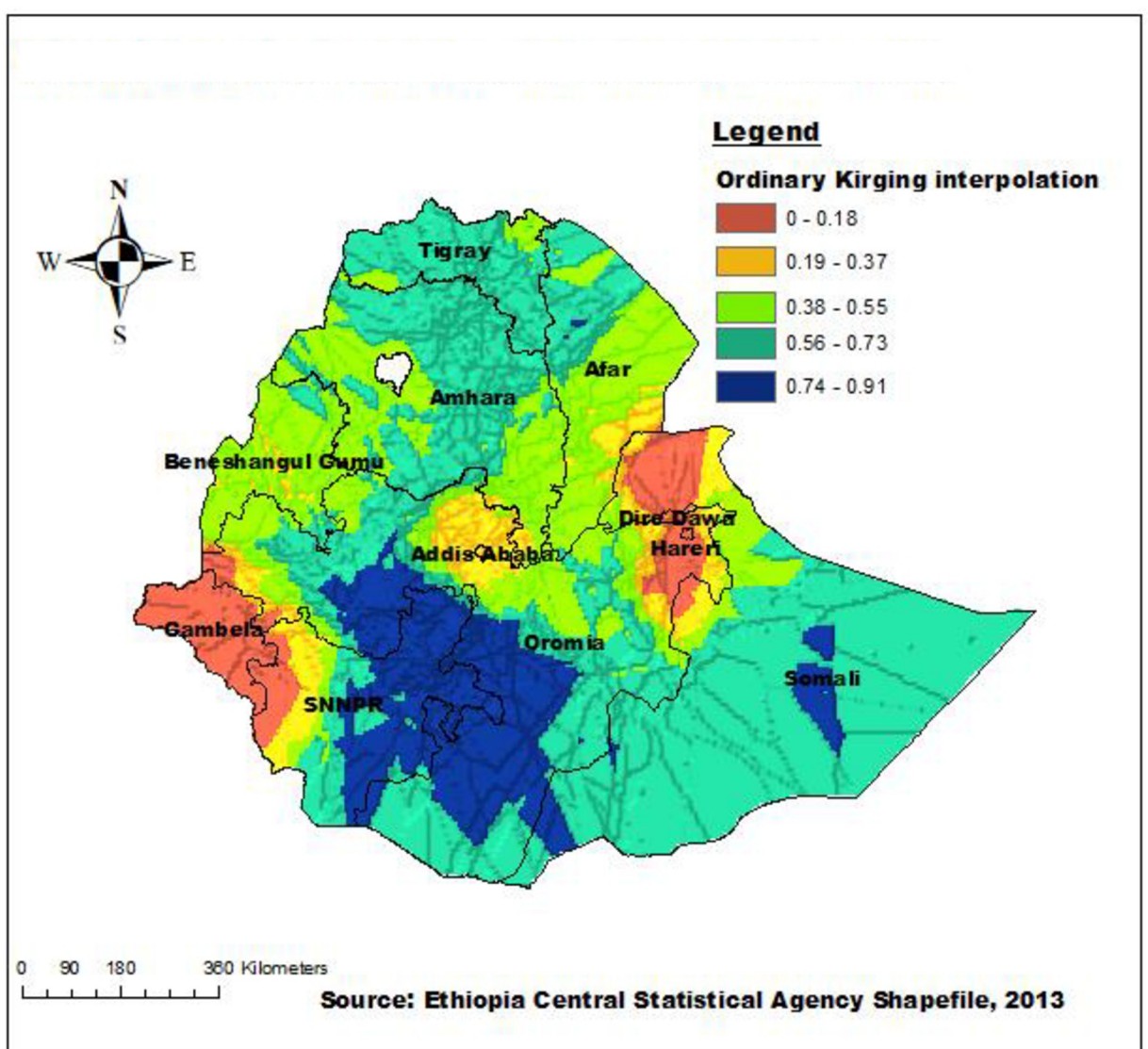

**Fig 4. Interpolated distribution of late initiation of first ANC visits in Ethiopia, 2019.**

## Random effect and model comparison

The Intraclass correlation coefficient (ICC) in the null model was (0.189), which means that 18.9% of the variability of late initiation of first ANC was due to the differences between clusters or unobserved factors at the community level. This indicates that the multilevel logistic regression model is best to estimate the predictors of late initiation of the first ANC visits among reproductive-age women than single-level logistic regression. Model four has a high log likelihood ratio (-991.9) and low deviance (1,983.8) as compared to the other models. Thus, model 4 is the best-fitted model. Therefore, all interpretations and reports were made based on this model. In addition, the median odds ratio (MOR) in all models was greater than one noting that there is a variation in the late initiation of first ANC visits among reproductive-age women between community levels. The value of MOR (2.39) in the null model depicts that there was a variation of late initiation of first ANC visit between clusters when we

**Table 3. Random effect and model comparison for predictors of late initiation of first antenatal care visit among reproductive-age women in Ethiopia, 2019.**

| Parameter | Null (model I) | Model II | Model III | Model IV |
|---|---|---|---|---|
| ICC | 18.9% | 16.2% | 13.4% | 9.6% |
| Variance | 0.86 (0.66, 1.21) | 0.35 (0.19, 0.69) | 0.33 (0.21, 0.52) | 0.16 (0.05, 0.49) |
| MOR | 2.39 | 1.53 | 1.48 | 1.03 |
| PCV | Reference | 59.3% | 61.6% | 81.4% |
| Model fitness | | | | |
| Log likelihood ratio (LLR) | -1813.8 | -1003.6 | -1784.6 | -991.9 |
| Deviance (-2LLR) | 3,627.6 | 2,007.2 | 3,569.2 | 1,983.8 |

ICC: Intraclass correlation coefficient, MOR: Median Odds ratio, PCV: proportional change in variance

randomly pick women from two clusters, women from clusters that have high rates of late initiation of first ANC visit were 2.39 times more likely to attend first ANC visits lately as compared to women from the low rates of late first ANC visit utilization cluster. The higher proportional change in variance (PCV) value in the fourth model showed that about 81.4% of the variability of late initiation of first ANC visits was explained by both the individual-level and community-level predictors (Table 3).

## Predictors of late initiation of first ANC visit among reproductive-age women in Ethiopia, 2019

As depicted in the final model (model 4), both the individual and community-level predictors were added in the multi-level analysis, maternal educational status, household wealth status, sex of household head, family size, residence, and region were identified as significant predictors of late initiation of the first ANC visits in Ethiopia. The odds of late initiation of first ANC visits were 38% (AOR = 0.62: 95%CI; 0.26–0.94) less likely among women who attended higher education as compared to those women who didn't attend formal education. Women who belong to the household wealth status of medium, richer, and richest were 58% (AOR = 0.42: 95%CI; 0.15–0.83), 69% (AOR = 0.31: 95%CI; 0.15–0.62), and 75% (AOR = 0.25: 95%CI; 0.13–0.50) less likely to have late first ANC visit compared to women who belong to poorest wealth status, respectively. The odds of late initiation of the first ANC visit were two times (AOR = 1.86: 95%CI; 1.21–2.85) more likely among women whose households were headed by a male as compared to their counterparts. Likewise, the odds of late initiation of first ANC visits were 1.5 times (AOR = 1.46: 95%CI; 1.13–2.28) more likely among women who had ≥ 5 family sizes as compared to those women who have had less than five family sizes. Rural dweller women had 1.5 times (AOR = 1.51: 95%CI; 1.05–5.63) more likely to attend their first ANC visits lately than women who lived in urban. Women who lived in the SNNPRs (AOR = 3.13: 95%CI; 1.12–8.77) and Oromia regions (AOR = 2.98: 95%CI; 1.26–7.08) were three times more likely to have late first ANC visits as compared with women living in Addis Ababa, the capital city of Ethiopia (Table 4).

## Discussion

Despite the proven benefits of early initiation of first ANC visits to improve maternal and neonatal health outcomes through early detection and prevention of risks during pregnancy [44, 45], shreds of evidence showed that numbers of women initiating their first ANC visit lately in Ethiopia [46]. Thus, this study aimed to give up-to-date information about the hotspot

**Table 4. Multilevel logistic regression analysis of the predictors of late initiation of first ANC visits among reproductive-age women in Ethiopia, 2019.**

| Variable | Category | Null model I | Model II AOR (95%CI) | Model III AOR (95%CI) | Model VI AOR (95%CI) |
|---|---|---|---|---|---|
| Maternal age (Years) | 15–19 | - | 1 | - | 1 |
| | 20–29 | - | 0.75 (0.33, 1.72) | - | 0.77 (0.34, 1.73) |
| | 30–39 | - | 0.56 (0.24, 1.31) | - | 0.58 (0.25, 1.34) |
| | 40–49 | - | 0.74 (0.19, 2.96) | - | 0.73 (0.19, 2.76) |
| Marital status | Married | - | 1 | - | 1 |
| | Unmarried | - | 0.98 (0.48, 1.99) | - | 0.99 (0.49, 2.00) |
| Religion | Orthodox | - | 1 | - | 1 |
| | Muslim | - | 1.11 (0.67, 1.85) | - | 0.86 (0.46, 1.62) |
| | Protestant | - | 2.16 (1.23, 3.76) | - | 1.28 (0.64, 2.56) |
| | Others [c] | - | 1.99 (0.61, 6.48) | - | 1.19 (0.35, 4.10) |
| Educational status | No education | - | 1 | - | 1 |
| | Primary | - | 0.90 (0.61, 1.34) | - | 0.86 (0.58, 1.29) |
| | Secondary | - | 1.31 (0.67, 2.56) | - | 1.31 (0.66, 2.59) |
| | Higher | - | 0.55 (0.24, 1.27) | - | 0.62 (0.26, 0.94)* |
| Household wealth status | Poorest | - | 1 | - | 1 |
| | Poorer | - | 0.53 (0.27, 1.07) | - | 0.48 (0.24, 1.01) |
| | Middle | - | 0.47 (0.25, 0.91) | - | 0.42 (0.22, 0.83)* |
| | Richer | - | 0.33 (0.16, 0.65) | - | 0.31 (0.15, 0.62)* |
| | Richest | - | 0.20 (0.11, 0.36) | - | 0.25 (0.13, 0.50)* |
| Birth interval (in months) | < 24 | - | 1 | - | 1 |
| | 24–36 | - | 1.57 (0.86, 2.88) | - | 1.63 (0.89, 3.01) |
| | >36 | - | 1.73 (0.91, 3.29) | - | 1.77 (0.93, 3.37) |
| Contraceptive method used | No-method | - | 1 | - | 1 |
| | Modern | - | 0.80 (0.56, 1.16) | - | 0.79 (0.55, 1.14) |
| | Traditional | - | 2.48 (0.44, 13.94) | - | 2.41 (0.38, 7.06) |
| Sex of household head | Female | - | 1 | - | 1 |
| | Male | - | 1.70 (1.11, 2.59) | - | 1.86 (1.21, 2.85)* |
| Family size | ≤ 4 | - | 1 | - | 1 |
| | ≥ 5 | - | 1.48 (0.96, 2.31) | - | 1.46 (1.13, 2.28)* |
| Community-level variables | | | | | |
| Residence | Urban | - | - | 1 | 1 |
| | Rural | - | - | 2.14 (1.42, 3.22) | 1.51 (1.05, 5.63)* |
| Region | Addis Ababa | - | - | 1 | 1 |
| | Afar | - | - | 1.58 (0.83, 3.01) | 0.99 (0.32, 3.01) |
| | Amhara | - | - | 1.60 (0.85, 3.04) | 1.2 (0.49, 2.95) |
| | Oromia | - | - | 2.36 (1.25, 4.43) | 2.98 (1.26, 7.08)* |
| | Somalia | | - | 3.14 (1.40, 7.03) | 1.49 (0.52, 4.31) |
| | B/Gumz | | - | 2.30 (1.20, 4.02) | 2.18 (0.89, 5.33) |
| | SNNPR | | - | 3.99 (1.91, 8.36) | 3.13 (1.12, 8.77)* |
| | Gambella | | - | 2.93 (1.65, 5.19) | 1.51 (0.55, 4.15) |
| | Harari | | - | 0.71 (0.39, 1.28) | 0.88 (0.35, 2.17) |
| | Tigray | | - | 1.69 (0.94, 3.07) | 1.06 (0.48, 2.34) |
| | Dire Dawa | | - | 0.65 (0.34, 1.31) | 0.46 (0.17, 1.24) |

*$p$-value < 0.05,

AOR: Adjusted odds ratio, CI: Confidence interval, 1: Reference,

[c] catholic or traditional religion follower,

SNNPR: South Nation Nationality peoples region, B/Gumiz: Benshangul-Gumz region

areas of late initiation of the first ANC visit and its predictors in Ethiopia using the 2019 Ethiopia Mini Demographic and Health Survey data.

This study showed that 62.6% (95% CI: 59.8–65.3) of reproductive-age women had started their first ANC visit lately. This result was consistent with studies done in Ethiopia [20, 47, 48]. However, the finding of the study was lower than those of previously conducted studies [32, 34, 38, 49, 50]. This discrepancy could be due to the difference in the gaps in the study period [34, 38, 50], the launching and strengthened functioning of the Health Extension Program (HEP) and the women's development army, and improving access to health care systems in the country. On the other hand, compared to other previously employed studies [36, 37, 51, 52], the current study found a higher prevalence of late initiation of the first ANC visit. The variation could be due to the difference in the study settings (in which the previous studies were done in a single health institution with small sample size) and study populations; in which most of the participants in the previous studies were urban dwellers whereas more than two-thirds (70.2%) of the mothers who participated in the current study were rural residents. So, the higher prevalence of late initiation of first ANC visits reported in this study might be explained by women who lived in rural areas are less likely to have a nearby health facility which in turn exposed them to other extra costs for transportation services as well as lack of availability of means of transportations. As a result, they fail to attain ANC services timely. In addition, the other possible justification might be the difference in the operational definition used to classify the outcome variable of early initiation of ANC visit. Most of the previous studies defined "late initiation of first ANC visit" as if a woman starts her first ANC visit after 16 weeks of gestations, but, we used 12 weeks as the lower cut of point to classify as late initiation of first ANC visits as recommended by WHO [53].

Our study found that the spatial distribution of late initiation of the first ANC visit was significantly varied across regions in Ethiopia. The highest rates of late initiation of the first ANC visit were highly clustered in the eastern parts of SNNPR, western Oromia, eastern and central parts of Tigray, central Amhara, and a few places in the Somalia and Afar regions. Whereas, Dire Dawa, Harari, and Gambella regions were identified as having low rates of the late start of the first ANC visit. This study was in agreement with a study done in Ethiopia [54]. It was also supported by the multilevel analysis in this study which revealed that women who lived in the SNNPRs and Oromia regional states were more likely to have a late first ANC visit as compared with women living in Addis Ababa. This geographical variation of late initiation of the first ANC visit might be due to the regional disparities in the availability of health care facilities, socioeconomic status, and demographic factors such as the pastoralist region.

The study also revealed that women who attended higher education were less likely to start their first ANC visits lately as compared to those women who didn't attend formal education. This result was supported by studies employed in Ethiopia [20, 34, 46, 55, 56], Myanmar [57], Ghana [52], Nigeria [58], Northern Uganda [59], and Sub-Saharan Africa [35]. This is explained by the fact that educated women are more economically independent, employed, have good levels of knowledge about the benefits of attending ANC visits, the appropriate timing when it is started, and the negative consequences related to delayed initiation of ANC visits than those women who did not attend formal education [60].

The odds of a late start of first ANC visits and household wealth status were negatively associated. Women who belong to the poorest household were more likely to start their ANC visits lately as compared to women who belong to the medium, richer, and richest household wealth status. The finding was in agreement with studies employed in Ethiopia [20, 56, 61], Ghana [52], Cameroon [62], and Sub-Saharan Africa [35]. Even though maternal health services (i.e. ANC, skilled birth attendants, postnatal care, immunizations, and family planning services) are given free of charge for all women in Ethiopia at government health facilities, services fees

at private health facilities, and non-services-related costs like transportation fees [63, 64] are unacceptably high. In addition, most women are obliged to experience a long waiting time to get the services, going a long distance to and from health facilities. Such types of indirect costs are correlated with the women's day-to-day life at which they might go to a farm, market, office, and other workplaces to gain money to cover their daily living. As a reason, women belonging to the poorest households will be more likely to have a late first ANC visit.

The current study demonstrates that the odds of the late start of first ANC visits were more likely among women whose households were headed by a male as compared to their counterparts. This finding was supported by other studies results in which husbands' permission affected the timing of the first ANC visit initiation [65–67]. Another study done by Mulat G., et al. revealed that the odds of using ANC visits were more likely among women who had autonomy in their healthcare decision-making as compared to their counterparts [68]. Thus, the information dissemination and education about the merit of women's autonomy and empowerment in all dimensions of life should be much strengthened, particularly in their own healthcare decision.

Consistent with study done in Ethiopia [34], and Rwanda [69], the odds of late initiation of the first ANC visit were more likely among women who had ≥ 5 family members as compared to those women who have had less than five family members. The possible justification might be those woman who belongs to a large family size are more likely to be prone to financial deficiency, spend more time on caring for their family, and perform all other household activities than taking care of their health.

In agreement with previous studies done in Africa [20, 34, 57, 61], this study identified that the odds of late initiation of the first ANC visits were more likely among women who lived in rural areas as compared to women who lived in urban areas. The possible justification could be due to women who resided in rural areas being exposed to inadequate availability and accessibility of health facilities, and fewer chances of getting health information as compared to women who lived in urban areas.

Our findings also depicted that women who lived in the SNNPRs and Oromia regional states were more likely to have late first ANC visits as compared with women living in Addis Ababa, the capital city of Ethiopia. The variation could be due to the difference in accessibility and availability of health facilities; in which participants from Addis Ababa are more socioeconomically developed, and have better accessibility, and availability of health facilities, including both public and private health services [70].

The clinical and public health implication of this study is to increase the rate of early initiation of first ANC visits and to decrease the fetomaternal complications as well as bad perinatal outcomes that occurred due to missed opportunities. The result of this study provides information about the spatial distribution of late initiation of the first ANC visit and its predictors. Thus, policymakers should give due emphasis to the identified hotspot areas. Also, giving special attention to women from large family sizes, women from rural areas, women with no formal education, women from a household headed by husbands, women from poorer household wealth status, and Oromia and SNNPRs regions residents could minimize the late initiation of first ANC visits among pregnant women.

## Strength and limitation

The first strength of this study was that the study used nationality representative data of 2019 EMDHS with large sample size, a high response rate, and high-quality data which will minimize a bias correlated to sampling and measurement. Secondly, the geospatial distribution of the late start of the first ANC visit was spatially autocorrelated. Moreover, we have used an

appropriate statistical approach, multilevel mixed-effect analysis, to estimate the cluster effect on the late start of first ANC visits. Yet, we would like to assure our reader that a few limitations needed to take into account. As a cross-sectional study, the exact cause-effect relationship between early initiation of ANC visits and its predictors doesn't exist, and recall bias might be introduced. The other limitation was the study failed to address some important predictors like distance to the health facility, pregnancy intention, media exposure and covered by health insurance, which may affect the timing of first ANC visits.

## Conclusion

In summary, a study demonstrated a higher prevalence of late initiation of first ANC visit in Ethiopia. A study also detected a clustered patterns of areas with high rates of late initiation of first ANC visit across regions the country. Local clusters of statistically significant areas with a high rate of late initiation of the first ANC visit were observed in the eastern parts of SNNPR, western Oromia, eastern and central parts of Tigray, central Amhara, and a few places in the Somalia and Afar regions of the country. Being a rural resident, attending higher education, having medium wealth status, richer wealth status, richest wealth status, having $\geq 5$ family size, the household headed by a male, living in SNNPRs, and Oromia region were statistically significant predictors of late initiation of first ANC visits. Thus, the Ethiopian government, the minister of health, and other concerned stakeholders consider giving due emphasis to those identified hotspot areas and these predictors when designing new policies/updating existing policies. In addition to the provision of ANC services freely for all pregnant women at public health facilities, governmental and non-governmental organizations should try their best to steadily increase the availability and accessibility of health facilities in rural areas to optimize the timely initiation of ANC uptake. Also, ANC services should be provided at all health facilities levels consistently, particularly in rural areas and identified hotspot regions. Furthermore, women's autonomy and empowerment in all dimensions of life, particularly in their own healthcare decision, should need to be given due emphasis. Lastly, the future researcher will consider a qualitative study to explore more unidentified individual, community, and facility-level factors.

## Acknowledgments

The authors acknowledge the Demographic and Health Surveys data center for allowing and permitting us to access the full data set.

## Author Contributions

**Conceptualization:** Gossa Fetene Abebe, Anteneh Messele Birhanu, Desalegn Girma.

**Data curation:** Gossa Fetene Abebe, Anteneh Messele Birhanu, Ashenafi Assefa Berchedi, Yilkal Negesse.

**Formal analysis:** Gossa Fetene Abebe, Anteneh Messele Birhanu, Dereje Alemayehu, Desalegn Girma, Yilkal Negesse.

**Methodology:** Gossa Fetene Abebe, Desalegn Girma, Ashenafi Assefa Berchedi, Yilkal Negesse.

**Resources:** Ashenafi Assefa Berchedi.

**Software:** Gossa Fetene Abebe, Yilkal Negesse.

**Supervision:** Gossa Fetene Abebe, Dereje Alemayehu, Ashenafi Assefa Berchedi, Yilkal Negesse.

**Validation:** Gossa Fetene Abebe, Dereje Alemayehu, Desalegn Girma, Ashenafi Assefa Berchedi.

**Visualization:** Gossa Fetene Abebe, Anteneh Messele Birhanu, Dereje Alemayehu, Desalegn Girma, Ashenafi Assefa Berchedi, Yilkal Negesse.

**Writing – original draft:** Gossa Fetene Abebe.

**Writing – review & editing:** Gossa Fetene Abebe, Anteneh Messele Birhanu, Dereje Alemayehu, Desalegn Girma, Ashenafi Assefa Berchedi, Yilkal Negesse.

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
