## [Decision Letter · Decision Letter 0]

28 Apr 2023

PONE-D-22-23818Spatial distribution, and multilevel analysis of the predictors of late initiation of first antenatal care visit among reproductive-age women in Ethiopia: Using the Ethiopia Mini Demographic and Health Survey 2019PLOS ONE

Dear Dr. Abebe,

Thank you for submitting your manuscript to PLOS ONE. After careful consideration, we feel that it has merit but does not fully meet PLOS ONE’s publication criteria as it currently stands. Therefore, we invite you to submit a revised version of the manuscript that addresses the points raised during the review process.

We look forward to receiving your revised manuscript.

Kind regards,

Gizachew Gobebo Mekebo

Academic Editor

PLOS ONE

3. Ethics statement only appears at the end of the manuscript:

Your ethics statement should only appear in the Methods section of your manuscript. If your ethics statement is written in any section besides the Methods, please move it to the Methods section and delete it from any other section. Please ensure that your ethics statement is included in your manuscript, as the ethics statement entered into the online submission form will not be published alongside your manuscript

5. We note that Figures 3 and 4 in your submission contain [map/satellite] images which may be copyrighted. All PLOS content is published under the Creative Commons Attribution License (CC BY 4.0), which means that the manuscript, images, and Supporting Information files will be freely available online, and any third party is permitted to access, download, copy, distribute, and use these materials in any way, even commercially, with proper attribution. For these reasons, we cannot publish previously copyrighted maps or satellite images created using proprietary data, such as Google software (Google Maps, Street View, and Earth). For more information, see our copyright guidelines: http://journals.plos.org/plosone/s/licenses-and-copyright.

a. You may seek permission from the original copyright holder of Figures 3 and 4 to publish the content specifically under the CC BY 4.0 license. 

Additional Editor Comments:

Your manuscript was reviewed by two reviewers. It is found that the research topic  and the work sound good, but it needs to be revised very well. Therefore, we invite you to  address the points raised of the reviewers and significantly revise the manuscript.

Reviewers' comments:

Reviewer's Responses to Questions

**Comments to the Author**

1. Is the manuscript technically sound, and do the data support the conclusions?

Reviewer #1: Yes

Reviewer #2: Yes

2. Has the statistical analysis been performed appropriately and rigorously? 

Reviewer #1: Yes

Reviewer #2: Yes

3. Have the authors made all data underlying the findings in their manuscript fully available?

Reviewer #1: Yes

Reviewer #2: Yes

4. Is the manuscript presented in an intelligible fashion and written in standard English?

Reviewer #1: Yes

Reviewer #2: Yes

5. Review Comments to the Author

Reviewer #1: Dear editorial team of Plos one, thank you very much for inviting me to review this paper. Keep in touch for the future. I am kindly asking you excuse for being late to review!!

Reviewer Comments for authors to the manuscript ID” ID PONE-D-22-23818 [EMID:65d7eebb58f743df]

”

Comment #1: In abstract section, you weighted your sample size, for which analysis you weighted your sample size?? Is there variation with respect to cluster or outcome among study areas?? Please clarify why you want to weight your sample and at what stage do weighting is important??

Comment#2: Intraclass correlation coefficient (ICC), median odds ratio (MOR), and proportional change in variance (PCV) statistics were computed to determine variations of late initiation of first the ANC visit across clusters. As I think ICC and PCV are used for variability assessment and MOR is used for prediction of risk factors. Do you have clear justification to use MOR for variability assessment of the outcome within and among cluster??

Comment # Still in abstract section, you used the Akaike’s Information Criteria (AIC) to compare the best-fitted model and a model with the smallest AIC value was selected. As I know for correlated and clustered data structure Likely hood ratio test is the best method to compare model fitness. AIC and BIC are used for models of single level and uncorrelated data structure. Please convince me if you have references. It is better if you use LRT or change in deviance.

Comment #4: living in SNNPRs (AOR = 3.13: 95%CI; 1.12 3 – 8.77), and Oromia region (AOR = 2.98: 95%CI; 1.26 – 7.08) were risk factor to late initiation of first ANC. What is your comparison group to declare them as risk factor??

Comment #5: In result section of your abstract, the spatial analysis showed that late initiation of first ANC visits was significantly varied across the country. You conducted one study with in one country, Ethiopia. How do you say among countries? Please use the appropriate word like regions, zones woredas…..

Comment #6 The maternal mortality ratio (MMR) in Ethiopia, one of the Sub-Saharan African countries, is 412 maternal deaths per 100,000 live births. Please remove the phrase one of Sub Saharan country and simple say Ethiopia because even the Lehman person knows Ethiopia is the Sub-Saharan country.

Comment #7: your study population is those mothers live in selected EAs and the study included all reproductive-age women found in the selected clusters. Please differentiate clearly what are your study and sample populations and what is your clustering unit??

Comment #8: EDHS study uses stratified multistage sampling method and to consider stages dropped down, it uses design effect or ICC to consider perceived variability among and within cluster on outcome. Have you done it in your case?? If so why not you mentioned??

Comment#9: Sample weight is performed to descriptive analysis, spatial, and to consider and balance sample size variability. Why you performed for Multilevel analyses??.

Comment #10: In the methods section, page 10 please give the name Null model which is appropriate for the first model, a model without predictors, was fitted to determine the extent of cluster variation in the late initiation of first ANC visits.

Comment #12: Both bivariable and multivariable analyses were computed. What is the name of that analysis?? Please mention it??

Comment #13: you have community level factors but you didn’t told us how you aggregated those factors with their cluster IDs?? Please tell us how you aggregated and managed community level factors??

Comment #14: In your analysis, there are only two community level variables ( Residence and Region). To get better estimate and predict the outcome as much as possible, it is better if you include Community level educational status and community level wealth index by aggregating individual level values with their respective cluster ids/number. Please try it and test the difference.

Comment #14: You mentioned in strength and limitation section, page 21 some important predictors like distance to the health facility, pregnancy intention, media exposure and CBHI which may affect the timing of first ANC visits were not addressed. What is the reason for not addressing these variables??

Comment #15: Conclusion section:

a, When you conclude your study, please tell whether the prevalence of the expected outcome is high, low or normal.

b, Mention factors separately ( positively and negatively associated)

These are my concerns for the authors.

Thanks a lot again for inviting!!

Reviewer #2: Spatial distribution, and multilevel analysis of the predictors of late initiation of first antenatal care visit among reproductive-age women in Ethiopia: Using the Ethiopia Mini Demographic and Health Survey 2019

My concern

The idea of the work is sounding great. But I have the following minor comments

• Modify the topic

• The abstract is too long and gives a brief recommendation relying on your key findings

• Remove outdated references and support them with more updated references

• Try to match your objectives, methods, results, and conclusions

Decision: After making all the necessary updates accept the article for publications

The paper is good but the title needs to be revised. And also please add more updated references.

6. PLOS authors have the option to publish the peer review history of their article (what does this mean?). If published, this will include your full peer review and any attached files.

Reviewer #1: No

Reviewer #2: No

---

## [Author Response · Author response to Decision Letter 0]

25 May 2023

Author response to Editor and reviewers

Dear Editor and Reviewers,

Thank you very much for your email dated 28th Apr 2023 incorporating the insight of the editor and reviewer’s comments. We, the authors, would like to express our gratitude to you for the insightful and constructive review that has led to the great improvement of our paper entitled “Spatial distribution, and predictors of late initiation of first antenatal care visit in Ethiopia: spatial and multi-level analysis”. We declared that all the data underlying the results presented in the study are publicly available from the Measure DHS website: http://www.dhsprogram.com. We have carefully reviewed the comments given by the editor, and reviewers and revised the manuscript accordingly. Our responses are given in a point-by-point manner for the editor, and reviewers’ comments using the Author’s response to reviewer form. If you have any concerns to be addressed, we are happy to consider them.

Best regards!

Version 1: PONE-D-22-23818

Date: 5/25/2023

Academic editor comments and respective author’s response 

Editor comment 1: Please ensure that your manuscript meets PLOS ONE's style requirements, including those for file naming. The PLOS ONE style templates can be found at https://journals.plos.org/plosone/s/file?id=wjVg/PLOSOne_formatting_sample_main_body.pdf and https://journals.plos.org/plosone/s/file?id=ba62/PLOSOne_ formatting_ sample_ title_authors_affiliations.pdf

Authors Response: Thanks very much for this comment. The whole part of the manuscript has been updated as per the PLOS ONE style templates. 

Editor comment 2. In your Data Availability statement, you have not specified where the minimal data set underlying the results described in your manuscript can be found. PLOS defines a study's minimal data set as the underlying data used to reach the conclusions drawn in the manuscript and any additional data required to replicate the reported study findings in their entirety. All PLOS journals require that the minimal data set be made fully available. For more information about our data policy, please see http://journals.plos.org/plosone/s/data-availability.

Authors Response: Thanks very much for this constructive comment. The comment has been accepted and revision has been made to the “Data availability statement” (See on the cover letter above).

Editor comment 3. Ethics statement only appears at the end of the manuscript:

Your ethics statement should only appear in the Methods section of your manuscript. If your ethics statement is written in any section besides the Methods, please move it to the Methods section and delete it from any other section. Please ensure that your ethics statement is included in your manuscript, as the ethics statement entered into the online submission form will not be published alongside your manuscript.

Authors Response: Thanks a lot, dear editor, for this insightful comment. The ethics statements appear only in the Methods section of the manuscript (see page 10, lines 1-6). 

Editor comment 4: Please include a separate caption for each figure in your manuscript.

Authors Response: Thanks very much dear editor for this helpful comment. A separate caption has been included for each figure. 

Editor comment 5: We note that Figures 3 and 4 in your submission contain [map/satellite] images which may be copyrighted. All PLOS content is published under the Creative Commons Attribution License (CC BY 4.0), which means that the manuscript, images, and Supporting Information files will be freely available online,……

Authors Response: Thanks very much for this insightful comment. The map we used in figure 3 and 4 are freely available at: https://africaopendata.org/dataset/ethiopia-shapefiles.

Reviewer #1 comments and an author response

Comments

Reviewer #1: Dear editorial team of Plos one, thank you very much for inviting me to review this paper. Keep in touch for the future. I am kindly asking you excuse for being late to review!!

Reviewer comments for authors to the manuscript ID” ID PONE-D-22-23818 [EMID:65d7eebb58f743df]”.

Comment #1: In abstract section, you weighted your sample size, for which analysis you weighted your sample size?? Is there variation with respect to cluster or outcome among study areas?? Please clarify why you want to weight your sample and at what stage do weighting is important.

Authors Response: Thanks very much, dear reviewer, for these questions. Before we did any statistical analysis, we weighted the data using sample weight (V005) for probability sampling and non-response to restore the representativeness of the survey and get reliable statistical estimates. There are variations between clusters with respect to the outcome variable (i.e. variation of late initiation of first the ANC visit across clusters). 

Comment#2: Intraclass correlation coefficient (ICC), median odds ratio (MOR), and proportional change in variance (PCV) statistics were computed to determine variations of late initiation of first the ANC visit across clusters. As I think ICC and PCV are used for variability assessment and MOR is used for prediction of risk factors. Do you have a clear justification to use MOR for variability assessment of the outcome within and among clusters??

Authors Response: Thanks very much for this insightful question. The MOR determines the variations of late initiation of the first ANC visit at the high-risk cluster (clusters having high rates of late initiation of first ANC visit) and low-risk cluster (clusters having low rates of late initiation of first ANC visit) when we randomly pick two women during data collection from two clusters. Thus, MOR can use to determine the variability of the outcome variable among two randomly selected individuals from two clusters, clusters having high risk and low risk (Merlo J. et.al, 2005).

Merlo J, Chaix B, Yang M, Lynch J, Råstam L: A brief conceptual tutorial of multilevel analysis in social epidemiology: linking the statistical concept of clustering to the idea of contextual phenomenon. Journal of Epidemiology & Community Health 2005, 59(6):443-449.

Comment # Still in the abstract section, you used the Akaike’s Information Criteria (AIC) to compare the best-fitted model and a model with the smallest AIC value was selected. As I know for correlated and clustered data structure Likely hood ratio test is the best method to compare model fitness. AIC and BIC are used for models of single-level and uncorrelated data structures. Please convince me if you have references. It is better if you use LRT or change in deviance.

Authors Response: Thanks very much, dear reviewer, for this constructive comment. We accept the comment and correction has been made in the model comparison criteria as directed (See page 9, lines 12 to 14, and page 14 lines 3-4 & table 3). 

Comment #4: living in SNNPRs (AOR = 3.13: 95%CI; 1.12 3 – 8.77), and Oromia region (AOR = 2.98: 95%CI; 1.26 – 7.08) were risk factors to late initiation of first ANC. What is your comparison group to declare them as risk factors??

Authors Response: Thanks very much, dear reviewer, for this question. We compared all regions with Addis Ababa city administration, the capital city of Ethiopia (See page 15, lines 14-16). 

Comment #5: In result section of your abstract, the spatial analysis showed that late initiation of first ANC visits was significantly varied across the country. You conducted one study within one country, Ethiopia. How do you say among countries? Please use the appropriate word like regions, zones woredas…..

Authors Response: Thanks very much, dear reviewer, for your insightful comment. We accept the comment and correction has been made accordingly (See page 2, lines 16-17). 

Comment #6: The maternal mortality ratio (MMR) in Ethiopia, one of the Sub-Saharan African countries, is 412 maternal deaths per 100,000 live births. Please remove the phrase one of Sub Saharan country and simple say Ethiopia because even the Lehman person knows Ethiopia is the Sub-Saharan country.

Authors Response: Thanks very much, dear reviewer, for your insightful comment. We accept the comment and correction has been made accordingly (See page 3, lines 9-10). 

Comment #7: your study population is those mothers live in selected EAs and the study included all reproductive-age women found in the selected clusters. Please differentiate clearly what are your study and sample populations and what is your clustering unit??

Authors Response: Thanks very much, dear reviewer, for your question and comment. All reproductive-age women (15 - 49 years) who gave birth in the five years preceding the survey and who had at least one ANC visits for their last child all over Ethiopia were the source population, whereas women who gave birth in the five years preceding the survey and who had at least one ANC visit for their last child and lived in the selected enumeration areas (EAs) were the study populations. Enumeration areas (EAs) were the clustering unit (See page 6, lines 15-24). 

Comment #8: EDHS study uses stratified multistage sampling method and to consider stages dropped down, it uses design effect or ICC to consider perceived variability among and within cluster on outcome. Have you done it in your case?? If so why not you mentioned??

Authors Response: Thanks very much, dear reviewer, for your suggestion and question. To determine the variation of the outcome variable within clusters and between clusters, we did the Intraclass correlation coefficient (ICC), and the result was reported under the sub-heading random effect and model comparison (See page 13, lines 24-25). 

Comment#9: Sample weight is performed to descriptive analysis, spatial, and to consider and balance sample size variability. Why you performed for Multilevel analyses??.

Authors Response: Thanks very much, dear reviewer, for your question. We weighted the data using the sample weight (V005) before any statistical analysis to restore the representativeness of the survey for unequal sample sizes across clusters and to tell the STATA to take into account the sampling design when calculating the standard errors to get reliable statistical estimates. Given that, we have one data set and we used this data set for all analysis. 

Comment #10: In the methods section, page 10 please give the name Null model which is appropriate for the first model, a model without predictors, was fitted to determine the extent of cluster variation in the late initiation of first ANC visits.

Authors Response: Thanks very much, dear reviewer, for this insightful comment. We accept the comment and corrected as directed (See page 9, line 3). 

Comment #12: Both bivariable and multivariable analyses were computed. What is the name of that analysis?? Please mention it??

Authors Response: Thanks very much, dear reviewer, for this suggestion and question. The full name of the analyses corrected as both bivariable and multivariable multi-level logistic regression analysis were computed (See page 9, lines 7-8). 

Comment #13: you have community level factors but you didn’t told us how you aggregated those factors with their cluster IDs?? Please tell us how you aggregated and managed community level factors??

Authors Response: Thanks very much, dear reviewer, for your suggestion as well as question. We have only two community level variables namely residence and region and we have no aggregated variables.

Comment #14: In your analysis, there are only two community level variables (Residence and Region). To get better estimate and predict the outcome as much as possible, it is better if you include Community level educational status and community level wealth index by aggregating individual level values with their respective cluster ids/number. Please try it and test the difference.

Authors Response: Thanks very much, dear reviewer, for your recommendation. In our study, we didn’t use aggregated variables as community level variables because we fear that, if we use those individual variables as aggregated community level variables, there is multicollinearity between the individual level variable and the aggregated individual level variables. Thus, to handle the effect of multicollinearity, we did not to use those aggregated variables. 

Comment #14: You mentioned in strength and limitation section, page 21 some important predictors like distance to the health facility, pregnancy intention, media exposure and CBHI which may affect the timing of first ANC visits were not addressed. What is the reason for not addressing these variables??

Authors Response: Thanks very much, dear reviewer, for your question. The 2019 Ethiopia Mini Demographic Health Survey failed to address these listed variables, meaning these listed variables were not collected in the 2019 EmDHS. 

Comment #15: Conclusion section:

a, When you conclude your study, please tell whether the prevalence of the expected outcome is high, low or normal.

Authors Response: Thanks very much, dear reviewer, for this insightful comment. We accept the comment and correction has been made as directed (See page 21, lines 9-10).

b, Mention factors separately ( positively and negatively associated).

Authors Response: Thanks very much, dear reviewer, for this insightful recommendation. We accept the recommendations and it was organized in that way (See starting from the last paragraph of page 18 to page 20). 

Reviewer #2: Spatial distribution, and multilevel analysis of the predictors of late initiation of first antenatal care visit among reproductive-age women in Ethiopia: Using the Ethiopia Mini Demographic and Health Survey 2019

My concern

The idea of the work is sounding great. But I have the following minor comments

• Modify the topic.

Authors Response: Thanks very much, dear reviewer, for your encouraging comments and suggestions. We accept the comment and correction has been taken as indicated (See the topic on page 1). 

•The abstract is too long and gives a brief recommendation relying on your key findings.

Authors Response: Thanks very much, dear reviewer, for your recommendations. We accept the recommendation and corrected as directed. 

• Remove outdated references and support them with more updated references.

Authors Response: Thanks very much, dear reviewer, for your recommendations. We accept the recommendation and corrected as directed. 

• Try to match your objectives, methods, results, and conclusions.

Authors Response: Thanks very much, dear reviewer, for your recommendations. We accept the recommendation and corrected as indicated.

---

## [Decision Letter · Decision Letter 1]

6 Jul 2023

Spatial distribution, and predictors of late initiation of first antenatal care visit in Ethiopia: spatial and multilevel analysis

PONE-D-22-23818R1

Dear Dr. Abebe,

We’re pleased to inform you that your manuscript has been judged scientifically suitable for publication and will be formally accepted for publication once it meets all outstanding technical requirements.

Kind regards,

Gizachew Gobebo Mekebo

Academic Editor

PLOS ONE

Additional Editor Comments (optional):

Reviewers' comments:

Reviewer's Responses to Questions

**Comments to the Author**

1. If the authors have adequately addressed your comments raised in a previous round of review and you feel that this manuscript is now acceptable for publication, you may indicate that here to bypass the “Comments to the Author” section, enter your conflict of interest statement in the “Confidential to Editor” section, and submit your "Accept" recommendation.

Reviewer #1: All comments have been addressed

Reviewer #2: All comments have been addressed

2. Is the manuscript technically sound, and do the data support the conclusions?

Reviewer #1: Yes

Reviewer #2: Yes

3. Has the statistical analysis been performed appropriately and rigorously? 

Reviewer #1: Yes

Reviewer #2: Yes

4. Have the authors made all data underlying the findings in their manuscript fully available?

Reviewer #1: Yes

Reviewer #2: Yes

5. Is the manuscript presented in an intelligible fashion and written in standard English?

Reviewer #1: Yes

Reviewer #2: Yes

6. Review Comments to the Author

Reviewer #1: (No Response)

Reviewer #2: Well addressed proceed with the current version. Thanks Well addressed proceed with the current version. Thanks Well addressed proceed with the current version. Thanks Well addressed proceed with the current version. Thanks Well addressed proceed with the current version. Thanks Well addressed proceed with the current version. Thanks

7. PLOS authors have the option to publish the peer review history of their article (what does this mean?). If published, this will include your full peer review and any attached files.

Reviewer #1: **Yes: **Yilma Chisha Dea, Assistant Professor of Biostatistics at Arba Minch University

Reviewer #2: No

---

## [Editor Report · Acceptance letter]

17 Jul 2023

PONE-D-22-23818R1 

Spatial distribution, and predictors of late initiation of first antenatal care visit in Ethiopia: spatial and multilevel analysis 

Dear Dr. Abebe:

I'm pleased to inform you that your manuscript has been deemed suitable for publication in PLOS ONE. Congratulations! Your manuscript is now with our production department. 

Kind regards, 

on behalf of

Assistant Professor Gizachew Gobebo Mekebo 

Academic Editor

PLOS ONE